# APM-YOLOv7 for Small-Target Water-Floating Garbage Detection Based on Multi-Scale Feature Adaptive Weighted Fusion

**DOI:** 10.3390/s24010050

**Published:** 2023-12-21

**Authors:** Zhanjun Jiang, Baijing Wu, Long Ma, Huawei Zhang, Jing Lian

**Affiliations:** School of Electronic and Information Engineering, Lanzhou Jiaotong University, Lanzhou 730070, China; jiangzhanjun@lzjtu.edu.cn (Z.J.); malong@mail.lzjtu.cn (L.M.); zhanghuawei@mail.lzjtu.cn (H.Z.); lian322scc@163.com (J.L.)

**Keywords:** water-floating garbage management, small-target detection, YOLOv7, river channel outline extraction, multi-scale gated attention

## Abstract

As affected by limited information and the complex background, the accuracy of small-target water-floating garbage detection is low. To increase the detection accuracy, in this research, a small-target detection method based on APM-YOLOv7 (the improved YOLOv7 with ACanny PConv-ELAN and MGA attention) is proposed. Firstly, the adaptive algorithm ACanny (adaptive Canny) for river channel outline extraction is proposed to extract the river channel information from the complex background, mitigating interference of the complex background and more accurately extracting the features of small-target water-floating garbage. Secondly, the lightweight partial convolution (PConv) is introduced, and the partial convolution-efficient layer aggregation network module (PConv-ELAN) is designed in the YOLOv7 network to improve the feature extraction capability of the model from morphologically variable water-floating garbage. Finally, after analyzing the limitations of the YOLOv7 network in small-target detection, a multi-scale gated attention for adaptive weight allocation (MGA) is put forward, which highlights features of small-target garbage and decreases missed detection probability. The experimental results showed that compared with the benchmark YOLOv7, the detection accuracy in the form of the mean Average Precision (mAP) of APM-YOLOv7 was improved by 7.02%, that of mmAP (mAP0.5:0.95) was improved by 3.91%, and Recall was improved by 11.82%, all of which meet the requirements of high-precision and real-time water-floating garbage detection and provide reliable reference for the intelligent management of water-floating garbage.

## 1. Introduction

With the rapid development of industrialization in coastal and river areas, the problem of water pollution caused by water-floating garbage is becoming increasingly serious, causing serious damage to the water environment, water resources, and aquatic ecological environment of the region. At present, the monitoring and management of water-floating garbage mostly rely on human resources, and regular and designated salvage and cleaning of floating garbage consume a large amount of human and material resources. At the same time, the efficiency is low, making it difficult to meet the needs of real-time management of water-floating garbage [1]. There is an urgent need to explore new technologies to change the traditional manual monitoring mode and enable more efficient monitoring and treatment of water-floating garbage. The emergence of computer vision technology has made it possible for the efficient monitoring and management of water-floating garbage. When using computer vision technology to monitor and manage water-floating garbage, the primary task is to accurately identify the garbage from the complex riverbank background. To that end, a stable and accurate detection and recognition algorithm model is needed to ensure more efficient and accurate monitoring and treatment of water-floating garbage.

In recent years, computer vision, especially deep learning, has developed rapidly and has been widely applied in various scene fields, bringing revolutionary changes to human production and life. For example, computer vision technology, especially deep learning, is widely used in the detection of water-floating objects on water surfaces, mainly relying on remote sensing images [2] to detect floating objects such as ships [3], aquatic organisms [4], and marine pollution [5] in order to ensure the safety of ship navigation and avoid collision accidents, protect the diversity and ecosystem of aquatic organisms, and protect water environment safety. Zhao et al. [6] proposed the YOLOv7 algorithm to enhance the detection of small targets at seas where significant interference from the sea environment during the detection process exists. In their study, on the basis of YOLOv7, a prediction head and SimAM (Simple Parameter Free Attention Module) were added to enhance the detection ability of small targets; since then, the algorithm has been applied to maritime search and rescue missions. Yang et al. [7] improved the detection accuracy of underwater robots by incorporating variable convolution in the backbone structure of YOLOv3 to fuse features between residual networks, thereby obtaining more global semantic information and improving the detection accuracy of underwater targets. Böer et al. [8] used YOLOv5 as the detection algorithm for underwater marine animals to automatically determine the abundance, type, and size of biological groups and created a detection classification dataset containing 10 types of marine animals, constructing a complete underwater biological detection and evaluation system. On the basis of the original YOLOv5s model, Liu et al. [9] changed the backbone network of mobileNet and introduced an attention mechanism to extract key features, improving the detection accuracy of marine debris. Ma et al. [10] proposed a deep learning-based marine oil spill monitoring method, which obtains oil spill risk information from sentinel-1 polarized synthetic aperture radar (PolSAR) images, and introduced a weighted probability model to evaluate coastal ecological risks, providing a new solution for the ecological risk assessment of coastal ecosystems.

In the field of water-floating garbage detection and management, many advanced universal object detection algorithms, such as YOLOv7, have achieved significant success [11,12]. However, compared to the object detection methods of other fields, the identification process of water-floating garbage faces the following problems: (1) from the perspective of environmental protection, water-floating garbage is located against a diverse and complex background, which makes it difficult to distinguish water-floating garbage; (2) from the perspective of object detection, small-target water-floating garbage contains limited features, and their appearance changes significantly during the floating process, resulting in the unsatisfactory detection performance of deep learning algorithms for water-floating garbage [13]. In addition, there is a lack of publicly available datasets in this field, which leads to a lack of sufficient data when training deep learning models, making it difficult to fully validate the algorithm and extend its generalization ability. The lack of relevant data in the long run results in limited research and advancement in this field [14]. Therefore, it is necessary to construct a comprehensive dataset based on the actual situation of water-floating garbage, explore new methods and optimization strategies, and improve the detection accuracy and generalization of deep learning technology in the field of water-floating garbage detection to better meet the needs of water-floating garbage management.

To address the above issues, the detection of water-floating garbage with morphological changes can be optimized through data preprocessing and data augmentation [15,16,17,18]. For example, Zhang [19] proposed an inland river channel outline extraction method based on the holistically nested edge detection (HED) network for irregular river channel outlines, which overcomes the challenges of weak robustness in extracting inland river channel outlines in complex field environments. Tang et al. [20] analyzed the characteristics of water surface images and used mean shift filtering to smooth out the interference of ripples, effectively suppressing uneven water surface illumination and water surface reflection. In addition, small targets have always been a focus and difficult point in the field of object detection due to limited information. The feature extraction ability of the network can be improved by deepening the network, integrating multi-scale features and attention mechanisms, and optimizing the recognition and detection performance [21]. Huang et al. [22] used a multi-scale fusion strategy to improve the detection accuracy of indoor small targets using the single shot detector (SSD) algorithm. Due to the non-stationary nature of sea clutter, feature extraction is unstable, which affects detection performance. Wu et al. [23] designed a small-target feature extraction method based on prior information. The method obtains prior information from historical echo data through kernel density estimation (KDE) and then uses the corresponding feature estimation method to extract features. The results show that this method can overcome the interference of sea clutter, improving the detection accuracy of small targets on the sea surface. To better suppress the interference of complex backgrounds and negative samples in images, Xu et al. [24] proposed a YOLOv5s-pp algorithm, which introduces the convolutional block attention module (CBAM) to improve the feature extraction ability for small targets. The attention mechanism enables the model to autonomously learn and weight different input features by calculating the correlation between them, making the model more sensitive to important regions in the image. The introduction of attention mechanisms in the model not only improves the target detection accuracy, reduces false positives and errors, and enhances model robustness, but also effectively simplifies the model structure and enhances its interpretability. The attention mechanism has the advantage of plug and play, and when combined with object detection models, it can achieve better detection results. Wen et al. [25] combined coordinate attention (CA) with the YOLOv5 model and proposed a modified YOLOv5s network with coordinate attention for underwater target detection (YOLOv5s-CA), which improved the detection performance of underwater targets. Zhang et al. [26] combined the YOLO algorithm with attention mechanisms and softer non-maximum suppression (NMS) algorithms as the detection and tracking algorithm for automatic water surface inspection vehicles. To improve the detection ability of small targets, Li et al. [27] incorporated the coordinate attention for efficient mobile network design strategically in the YOLOv7 framework, demonstrating certain advantages in floating-garbage recognition.

In summary, in order to achieve better detection results, it is necessary to comprehensively consider various scenarios and combine various optimization and improvement strategies in response to the difficulties and challenges in the detection of water-floating garbage. Therefore, by deeply analyzing the limitations of YOLOv7 for water-floating garbage detection, a small-target water-floating garbage detection method based on APM-YOLOv7 is proposed. The method proposed in this study overcomes the difficulties in morphologically variable small-target water-floating garbage detection and is capable of real-time detection of small-target water-floating garbage under complex backgrounds. It is concluded that APM-YOLOv7 provides technical support for the detection and management of small-target water-floating garbage detection under complex backgrounds.

The contributions of this study are summarized as follows:(1)We collected and made public a dataset for water-floating garbage detection in the Yellow River (YRDG) and proposed an adaptive Canny (ACanny) algorithm for river channel outline extraction, which could reduce the interference of complex riparian backgrounds.(2)The lightweight convolution PConv is introduced, and the PConv-ELAN is designed in the YOLOv7 network, which help reduce parameter redundancy and model size while acquiring more morphologically varied features of the water-floating garbage.(3)A multi-scale gated attention allocation network with adaptive weight allocation is proposed. By designing a multi-scale adaptive gated weight allocation network to enhance attention to small target areas, MGA is inserted into the backbone of YOLOv7 to improve the detection accuracy of water-floating garbage.

## 2. Materials and Methods

The YOLOv7 algorithm was proposed by Wang et al. [28], as shown in Figure 1. Compared with second-order target detection algorithms, such as Faster R-CNN, R-CNN, and Fast R-CNN, YOLOv7 adopts first-order regression and obtains the location and category of the corresponding target through convolutional operation at one time with fast detection speed and a better balance between speed and accuracy [29,30]. In addition, unlike first-order algorithms, such as YOLOv5, SSD, and CenterNet, YOLOv7 adopts faster convolutional operation and a more innovative network structure, making it popular in target detection [31]. However, there is still some room for improvement and optimization of YOLOv7 for small-target detection, especially for the detection of small-target floating garbage with frequent morphological changes. Therefore, in this study, we optimize and improve the YOLOv7 network with regard to backbone feature extraction capability and multi-scale feature fusion to enhance the detection performance.

As shown in Figure 1, the YOLOv7 network consists of four structures, namely Input, Backbone, Neck, and Head. The Input processes the input image, enhanced by Mosaic and Mix-up data, which is then input into the Backbone with adaptive filling and anchor frame preprocessing. The Backbone consists of a number of CBS modules; MP and ELAN structures extract the features of the input image, and the feature extraction is achieved through the superposition of convolution and jump connection. The Neck fuses the extracted features; then, the three detection heads of the Head structure output information about the category and location of the water-floating garbage.

In the following, we will mainly analyze the limitations of YOLOv7 network in the scenario of small-target water-floating garbage detection with large morphological changes during the drifting process. Improvement and optimization strategies are proposed to deal with actual floating-garbage scenarios.

### 2.1. Limitations of Small-Target Water-Floating Garbage Feature Extraction

Small-target detection has always been an important, while difficult, issue in target detection, in that small targets contain fewer pixels, and fewer effective features can be extracted. In addition, in terms of the network design, the deeper network structure and the convolution of larger step size commonly lead to the loss of small-target features [32,33,34].

Therefore, in order to improve small-target detection, researchers have optimized and improved multi-scale feature extraction methods such as contextual feature information fusion, a priori frame setting, the intersection-parallel ratio matching strategy, the non-great suppression method, loss function optimization, and the generative adversarial network and target feature extraction network structure, with considerable achievements having been made. However, compared with other small-target detection applications, small-target water-floating garbage detection faces many challenges. For example, small targets are easily merged with the background, and their colors are so similar to the color of waves, especially in the turbid Yellow River basin, that sometimes it is hard to identity small-target water-floating garbage, as shown in Figure 2.

Figure 3 shows the output feature maps of different convolutional layers of YOLOv7. To solve the feature loss problem of the network in extracting small targets, the small targets in the images in the VOC2007 dataset [35] and the water-floating garbage dataset are selected for analysis. It can be seen that feature loss occurs while YOLOv7 extracts features of small targets. In addition, in the water-floating garbage images, the features are still present after the fourth and eighth layers of convolutional feature extraction, but the features of water-floating metallic garbage are lost in the ninth layer. Similarly, in the VOC dataset, after the fourth and fifth layers of convolutional feature extraction, the features of small targets gradually disappear, and some of the features are lost in the sixth layer.

There are two reasons accounting for feature loss in the convolution process for small targets. In the convolution process, YOLOv7 uses several layers of convolution and pooling layers to achieve feature extraction, and the convolution is computed as follows:(1)$$F_{\mathrm{out}}=(F_{in}-K+2P)/s+1$$
where F represents the feature size, K denotes the convolution kernel size, P is the padding value, and s stands for the convolution step. When the convolution kernel size is larger than the target size and the convolution step is too large, the features contained in the small-target region are gradually reduced during the sliding convolution process. In YOLOv7, the maximum pooling is calculated as
(2)$$F_{\max\mathrm{pool}}\left(x,y\right)=\max\left(F_{in}\left[xs:x\left(s+h\right),ys:y\left(s+w\right)\right]\right)$$
where $\left(x,y\right)$ is the input image pixel point, and h and w are the sliding window size on the length and width, respectively. The pooling layer mainly reduces the dimensionality of the input data, which improves the inference speed and interpretability of the model. However, the pooling operation leads to a large amount of feature loss as well. In order to compensate for this feature loss, this study proposes a multi-scale gated attention MGA with adaptive weight assignment, which uses multi-scale convolution to extract features and utilizes the “gating” idea to adaptively assign weights to realize the weighted fusion of multi-scale features, thereby highlighting the features of small targets.

### 2.2. Limitations of Feature Extraction for Morphologically Variable Water-Floating Garbage

As shown in Figure 4, as it drifts, the water-floating garbage experiences enormous changes in appearance and morphology, caused by drifting, tumbling, and rotating.

In order to extract more features of water-floating garbage with morphological changes, as inspired by Faster Net (Faster Neural Networks), PConv [36] is introduced into YOLOv7 to replace the 3 × 3 convolution in the ELAN structure, for the purpose of reducing the redundant parameters while combining the channel and spatial features, and extracting more invariant features so as to improve detection accuracy.

## 3. Methods

The overall process of APM-YOLOv7 for small-target water-floating garbage proposed in this article is shown in Figure 5. Firstly, the Yellow River water-floating garbage dataset (YRDG) was input and processed by the ACanny algorithm to output the extracted image of the river channel outline. This preprocessing can reduce the interference of complex backgrounds in detection accuracy. Then, an improved model of PConv-ELAN and MGA (APM YOLOv7) was added to the backbone of YOLOv7 to train and test the images processed by the ACanny algorithm, which then output the results of water-floating garbage detection. APM YOLOv7 can enhance the feature extraction for small-target water-floating garbage and improve detection accuracy.

### 3.1. Canny Edge Detection Algorithm

The Canny edge detection algorithm comprises four main components [37].

Canny edge detection algorithm

The Canny edge detection algorithm preprocesses the image using gaussian smoothing filter to retain the details and edge information in the image while performing the denoising operation. In addition, gaussion smoothing filter improves the contrast of the image and enhances the edge features of the target. For the input image f(x,y), gaussian smoothing filter is processed as follows:(3)$$G(x,y)=\frac{1}{2\pi \sigma^{2}}\exp\left(-\frac{x^{2}+y^{2}}{2\sigma^{2}}\right)$$
(4)S(x,y)=G(x,y)×f(x,y)
where G(x,y) is the Gaussian filter function, S(x,y) represents the filtered image, σ is the coefficient of the Gaussian filter, and σ denotes the convolution operation.

2.Calculation of gradient magnitude and gradient direction

Canny uses first-order derivatives to compute the gradient and gradient direction. A(x,y) is the derivative of the horizontal direction x, and B(x,y) is the derivative of the vertical direction. The gradient magnitude value C(x,y) of the original image f(x,y) is computed to determine the direction of the function image θ(x,y).
(5)A(x,y)=S(x+1,y)−S(x,y)+S(x,y)−S(x−1,y)
(6)B(x,y)=S(x,y+1)−S(x,y)+S(x,y)−S(x,y−1)
(7)$$C(x,y)=\sqrt{A{\left(x,y\right)}^{2}+B{\left(x,y\right)}^{2}}$$
(8)$$\theta (x,y)=\arctan\left(\frac{B(x,y)}{A(x,y)}\right)$$
where C(x,y) and θ(x,y) are the gradient magnitude and angle values, respectively.

3.Edge refinement

Canny uses non-extremely large value suppression to refine the target edge to eliminate the spurious points accompanying the edge. After calculation of the gradient and gradient direction, it looks for the maximum point in the gradient amplitude and uses it as the edge point of the target. It is worth noting that not all gradient maximum points are edge points. Non-maximum suppression refers to further distinguishing these maximum points based on the gradient direction so as to determine the true edge points. The filtered gradient amplitude values C(x,y) are calculated as follows:(9)$$C(x,y)=\left\{\begin{array}{l} 0,ifC(x,y)>C(x-1,y-1)\&C(x,y)>C(x+1,y+1)\\ 1,else \end{array}\right.$$

The judgment is based on the fact that if the gradient value of the center pixel C(x,y) is greater than the gradient values of C(x−1,y−1) and C(x+1,y+1) in the same direction, then the point is an edge point.

4.Edge point screening and connection

Thresholding the edge connection of the image after edge refinement is performed as follows: calculate the gradient magnitude of the pixel point, and if the gradient magnitude of the pixel point is greater than the high threshold, it is determined to be an edge point; if the gradient magnitude of the pixel point is less than the low threshold, it is determined to be a non-edge point; if the gradient magnitude of the pixel point is in between the high and low thresholds, further judgement is conducted regarding whether there are any points greater than the high threshold in its eight-neighborhood. If the answer is yes, then it is determined to be an edge point; otherwise, it is a non-edge point.

The Canny algorithm requires manual parameter setting, resulting in weak generalizability. Moreover, the threshold set based on empirical values will, to a large extent, determine whether edge points are present, and local feature information cannot be taken into account, resulting in certain errors in edge determination.

### 3.2. Adaptive Canny (ACanny) Algorithm for River Channel Outline Extraction

To overcome the shortcomings of the Canny algorithm, the ACanny algorithm is proposed, which has been improved through two main steps: denoising filtering and edge point filtering, and connection. Figure 6 shows the overall flowchart of the ACanny algorithm, and the detailed calculation process is as follows:

Step 1: Adaptively obtain Gaussian filter kernels of different scales and apply multi-scale Gaussian filtering to the input dataset to remove the noise interference of different scales.

Step 2: Calculate the grayscale gradient and gradient direction of the input image at 0, 45, 90, and 135 degrees, and obtain all edge points of the river channel outline in the image.

Step 3: Perform edge refinement on all edge points obtained in step 2, and select the true edge points from all river channel outline edge points.

Step 4: Use the concave packet algorithm to connect the edge points of the river channel outline to form a connected edge area, and set the pixel value of the image within this connected area to 0.

Step 5: Multiply the image obtained in Step 4 with the input image to obtain the extracted image of the river channel outline.

Adaptive multi-scale denoising filtering

Noise of different scales exists in the water-floating garbage dataset, and it cannot be effectively removed if a single structure of filter kernel is used for filtering. In order to achieve a better denoising effect, this study combines Gaussian filtering and the eight-domain correlation property, utilizing the autocorrelation matrix to adaptively select Gaussian filter kernels of sizes 3 × 3 and 5 × 5 to perform multi-scale filtering on the input image. Equation (4) is improved as
(10)$$S\left(x,y\right)=G_{3\times 3}\left(x,y\right)\times f\left(x,y\right)⊙G_{5\times 5}\left(x,y\right)\times f\left(x,y\right)$$
where $G_{3\times 3}\left(x,y\right)$ and $G_{5\times 5}\left(x,y\right)$ are Gaussian filter kernels for the covariance matrix of the input image and ⊙ is the Hadamard product.

2.Calculation of gradient magnitude and gradient direction

Canny only considers the gradient in the horizontal and vertical directions when counting the pixel gradient; it does not consider the gradient information in the tilt direction, which leads to low accuracy in gradient calculation and an insufficient number of screened extreme value points, resulting in a high number of edge breakpoints. In order to reduce the breakpoints, this study adds the gradient components in the directions of 45° and 135° in turn, and determines the final gradient magnitude and the final gradient direction according to the weighted sum of distances. Equation (7) is improved as follows:(11)$$D_{45}\left(x,y\right)=S(x+1,y+1)-S(x,y)+S(x,y)-S(x-1,y-1)$$
(12)$$D_{45}\left(x,y\right)=S(x+1,y+1)-S(x,y)+S(x,y)-S(x-1,y-1)$$
(13)$$C(x,y)=\sqrt{A{\left(x,y\right)}^{2}+B{\left(x,y\right)}^{2}+D_{45}{\left(x,y\right)}^{2}+D_{135}{\left(x,y\right)}^{2}}$$
where $D_{45}\left(x,y\right)$ and $D_{45}\left(x,y\right)$ stand for the two newly added gradient components.

3.Adaptive edge point screening and connectivity

Canny’s high and low thresholds are set according to empirical values; this often requires a large amount of data to be determined experimentally, resulting in weak generalizability of the algorithm. In this study, we adaptively determine the double threshold according to Otsu [38].
(14)$$u=w_{0}u_{0}+w_{1}u_{1}$$
(15)$$t=w_{0}{\left(u_{0}-u\right)}^{2}+w_{1}{\left(u_{1}-u\right)}^{2}$$
where $w_{0}$ is the proportion of the riverbank foreground, $w_{1}$ is the proportion of the riverbank background points, $u_{0}$ is the mean gray value of the riverbank foreground, and $u_{1}$ is the mean gray value of the riverbank background as the global gray value. t stands for the maximum threshold, and the minimum value is assigned according to the ratio of maximum and minimum thresholds. The double thresholds of the Canny algorithm are set to 0.075 and 0.175 according to the researchers’ experience that the ratio of high and low thresholds is 7:3. Through adaptive processing, more accurate edge point calculation can be achieved.

4.Edge connection based on concave packet algorithm

After implementing the image edge detection using the Canny operator, a non-closed region is formed. Then, the concave packet algorithm can be used to highlight the edge features and generate smoother edges of the river channel outline. The idea is that for a given target pixel point, whether it is located on the edge is determined by calculating the convex packet formed by its surrounding pixel points. The edge connection based on the concave packet algorithm includes the following steps:(1)Edge determination: First, calculate the adjacent points around each pixel, and calculate the convex hull formed by these adjacent points. If the convex envelope contains that pixel, the pixel is considered an edge pixel.(2)Edge connection: Connect adjacent edge pixels to form a complete edge area. Specifically, through the region growth algorithm, start from a certain pixel point, repeat steps (1) and (2) until the maximum diameter is reached, and then stop the iteration.

As shown in Figure 7, compared with the Canny edge detection algorithm, ACanny as proposed in this study can extract the river channel outline more accurately and reduce the interference caused by complex river backgrounds. Through the river extraction preprocessing method, the river is segmented and extracted from the complex background, which makes the detection and recognition network capable of more accurately extracting the features of water-floating garbage, thus improving the detection accuracy.

### 3.3. APM-YOLOv7 Algorithm

The APM-YOLOv7 network designed in this study is shown in Figure 8. In order to improve the detection performance of the YOLOv7 network in the small-target water-floating garbage detection scenario, the YOLOv7 network was optimized and improved in terms of the following two aspects, as motivated by the limitation analysis in Section 2.

#### 3.3.1. PConv-ELAN Backbone Design

The ELAN module is an efficient layer aggregation network that is designed in such a way that more features are extracted by the network through jump connections and successive convolutions to ensure that it is more robust. As shown in Figure 9, the first branch of the ELAN module passes through a 1 × 1 Bconv to perform the channel number change. The second one firstly passes through a 1 × 1 Bconv and then passes through four 3 × 3 Bconvs to realize the feature extraction of continuous convolution; it finally passes through the feature concate fusion output. In this study, in order to extract more features of morphologically variable water-floating garbage, a lightweight convolutional layer, PConv [36], was used to design the PConv-ELAN network.

As can be seen in Figure 10, the conventional 3 × 3 Bconv is replaced by PConv, which is a new partial convolution that can reduce redundant computation and parameters and extract the spatial features of the target more efficiently. PConv firstly performs the split decomposition of the features of the input H × W × C according to channel C. In this study, C is divided into four copies, and the first three copies of the features go through the 3 × 3 PConv for feature extraction. The results obtained in last step are then directly spliced with the last copy, which is finally integrated into a layer of 1 × 1 convolution to complete the feature extraction.

The number of parameters of the conventional convolution is calculated as follows:(16)$$F_{\mathrm{convolution}}=H\times W\times C_{in}/g\times C_{out}$$

Conventional convolution is computed with *g* as the group size, and g=1, so Equation (16) can be simplified as follows:(17)$$F_{\mathrm{convolution}}=H\times W\times C^{2}$$

The parameters of PConv are computed as follows:(18)$$F_{PC\mathrm{onv}}=H\times W\times {\left(\frac{C}{4}\right)}^{2}$$

The number of parameters of PConv is only 1/16 of that of conventional convolution, and the PConv superposition can realize fast feature extraction with less parameter redundancy.

#### 3.3.2. Multi-Scale Gated Attention MGA with Adaptive Weight Assignment

In order to improve small-target feature extraction, this study proposes a multi-scale gated attention named MGA with adaptive weight allocation. The location of MGA is shown in Figure 7. For the feature extraction of small-target water-floating garbage, considering that the small targets in water-floating garbage may become blurred or even lose corresponding features in the deep feature maps, the MGA attention is added after the second layer of PConv-ELAN.

As shown in Figure 11 inspired by pyramid split multi-scale self-attention (PSA) [39] and the Omni-scale Network (OSNet) [40], a multi-scale gated attention MGA with adaptive weight assignment is designed. PSA firstly goes through the SPC module to realize the fusion of multi-scale feature extraction; then, the channel attention is used to realize feature weighting, and finally, the features are multiplied to output the final features. SPC uses four kinds of convolution kernels—3, 5, 7, and 9—and four kinds of convolution steps—2, 4, 8, and 16—to perform multi-scale feature extraction. OSNet is a new convolutional structure that can extract the full-scale features and better match the features. It is applied to target tracking and target re-identification by designing different convolutions and depth-separable convolutional (Light Conv) overlays to ensure that the network extracts features at different scales while consuming very little parameter computation. It should be noted that OSNet generates channel dynamic weights from the unified aggregation gate (AG), which weights the multi-scale features obtained by fusion. This novel gate design provides the OSNet network with great flexibility to realize multi-scale feature fusion with different weights according to different input images.

Inspired by the above two networks, MGA attention was proposed for small-target feature extraction, as shown in Figure 12. By designing different scales of depth-separable convolution (Lite Conv) for feature extraction, different weight sizes of features were output after AG feature sharing, and finally, the feature map after weighted feature fusion was obtained.

MGA represents the input feature $F_{in}$ extracted by four layers of Lite Conv, as follows:(19)$$F_{j}=\mathrm{LiteConv}\left(K_{j},g_{j}\right)\left(F_{in}\right)j=1,2,3,4$$
where j is the order of separable convolution, K is the convolution kernel size, and g is the group size. After the multi-scale separable convolution for feature extraction, the AG dynamically assigns weights to the feature maps extracted according to different scales and finally outputs the MGA processed maps with weighted summation.
(20)$$F_{\mathrm{MGA}}=\sum_{j=1}^{4}F_{j}⊙w_{\mathrm{AG}}$$
where ⊙ is the Hadamard product, and $w_{\mathrm{AG}}$ is the weight size calculated by AG.

## 4. Experimental Results and Discussion

### 4.1. Dataset

The dataset is a self-collected dataset based on the research results on intelligent identification and early warning technology for garbage detection within the river and lake areas in Gansu Province. The Yellow River, as a representative inland river, is characterized by fast flow velocity, low visibility, a complex riverbank environment, and illumination by waves. As shown in the captured images, the garbage floating on its surface is within a complex background with different-sized targets, and small targets account for a large proportion. Therefore, such an environment is typical and suitable for experimental research. The sources of water-floating garbage are diverse. In order to understand their distributional characteristics and types, we selected three representative areas for data collection: the tributary of the Yellow River, the main body of the Yellow River, and the reaches of residential areas along the river bank. Data collection was conducted in sunny and rainy days. These areas can fully reflect the distribution pattern of water-floating garbage in different water environments and weather conditions, which helps us to have a deeper understanding of the formation and accumulation of water-floating garbage.

The YRDG dataset has a total of 3807 images, with an image size of 640 × 640, and 84% of the targets are small targets. The dataset was divided into the test set (381 images), validation set (343 images), and training set (3083 images). Seven categories of water-floating garbage were included in the dataset: plastic, paper, glass, metal, fabric/fiber, natural garbage, and others. The quantity of each category and the large, medium, and small targets of each category are shown in Figure 13 and Table 1.

### 4.2. Experimental Environment and Evaluation Indicators

The experiments were conducted using the deep learning framework Pytorch, in Python 3.7.6, CUDA 10.0, and the GPU was NVIDIA GeForce RTX 3060 for single card training. In the training phase, pre-trained weights were used, the batch size was set to 4, the epoch was 200, the initialized learning rate was set to 0.01, and the weight decay coefficient was 0.000005.

In terms of detection accuracy, AP (accuracy of each category), mAP (average accuracy of each category), mmAP (detection accuracy under different IOUs, IoU = 0.5:0.95), and Recall (recall) were selected and used in this study. For the four evaluation indicators, the larger the value, the higher the detection accuracy and stability of the algorithm model. In addition to detection accuracy, model size and real-time detection performance are also important for detecting water-floating garbage. Therefore, Model Size and FPS (frames per second) were chosen as evaluation indicators. Model Size usually refers to the structure and number of parameters of the model. Generally speaking, the smaller the model, the smaller the number of parameters and the lower the computational complexity, which can result in the completion of calculations in a shorter time and accelerate the running speed of the model. FPS is usually used to measure the efficiency of object detection algorithms, reflecting the detection speed of the algorithm on a given hardware platform. In general, the higher the FPS value, the faster the detection speed of the detection algorithm.

### 4.3. Experimental Analysis

In order to objectively and accurately evaluate the comprehensive performance of the proposed model in this study in terms of detection accuracy, model size, computational complexity, and detection speed, we compared the performance of our model with seven deep learning target detection algorithms, including SSD [41], YOLOv5 [42], M2det [43], YOLOv7, Faster R-CNN (VGG-16) [44], YOLOX [45], and YOLOv8 [46]. The results are shown in Table 2.

From Table 2, it can be seen that the detection performance of the Faster R-CNN and M2det algorithms for small-target water-floating garbage are not satisfactory, with FPS values of 16.12 and 21.80, respectively, which cannot meet real-time detection requirements (the FPS index for real-time performance should be greater than or equal to 30). Among the eight comparative algorithms, YOLOv5s has the smallest model size, while YOLOX has the fastest detection speed and APM-YOLOv7 has the highest mAP, mmAP, and Recall metrics, accounting for 87.32%, 35.41%, and 84.53%, respectively. The FPS is 30.50 and the model size is 106.0 MB, which also possesses certain advantages and comparability. Compared with the latest object detection algorithms YOLOX and YOLOv8, the APM-YOLOv7 algorithm is more advantageous in detecting water-floating garbage in the YRDG dataset, and the detection performance is better. Compared with the benchmark YOLOv7, the proposed algorithm APM-YOLOv7 shows improvements in mAP by 7.02%, mmAP by 3.91%, and Recall by 11.82%. Due to the addition of MGA attention, FPS is increased by 2.07 and model size is increased by 34.6 MB, but it can still meet the needs of real-time detection.

From Figure 14, it can be seen that the APM-YOLOv7 algorithm possesses certain advantages in terms of detection accuracy AP for the detection of seven categories of water-floating garbage, with the highest AP for the plastic (83.14%), metal (89.12%), fabric/fiber (90.61%), and other (87.60%) categories. In the paper category, the SSD algorithm has the highest AP of 96.27%, and APM-YOLOv7 ranks third. In the glass category, the YOLOv5s algorithm has the highest AP value of 93.16%, and APM-YOLOv7 ranks fourth. In the nature category, the SSD algorithm has the highest AP value of 84.59%, and APM-YOLOv7 ranks third, indicating that the APM-YOLOv7 algorithm also exhibits certain accuracy and stability in detecting the paper, glass, and nature categories of garbage. Overall, the APM-YOLOv7 algorithm shows certain advantages in detecting different categories of water-floating garbage, including plastic, metal, fabric, fiber, paper, glass, etc. This indicates that the algorithm has good generalization ability and robustness and can be used to detect water-floating garbage in different scenarios.

Figure 15 shows the detection results of different algorithms. Although Faster R-CNN can effectively detect small-target water-floating garbage, there is a false detection in that two overlapping boxes occurred, as shown in the second and fourth images in Figure 15a. The YOLOv5s algorithm has missed detections, as shown in the first image in Figure 15b. There are many missed detections in the SSD algorithm, as shown in the third and fifth images in Figure 15c. YOLOv7 has leakage detections for small-target water-floating garbage and redundant overlapping frames, as shown in the third and fourth images in Figure 15d. The M2det algorithm has serious leakage detection for small-target water-floating garbage, as shown in Figure 15e, which indicates that the algorithm is not powerful in feature extraction for small-target water-floating garbage and that it needs to be further improved and optimized. Missed detections occurred in YOLOX, as shown in the second image in Figure 15f, with overlapping boxes used for detecting water-floating garbage, as shown in the third image in Figure 15f. YOLOv8 was accompanied with missed detections of small-target water-floating garbage, as shown in the fourth image in Figure 15g. APM-YOLOv7 did not show any missed or false detections, and its detection accuracy was higher on the same image. The above analysis suggests that satisfactory detection results have been obtained.

From the above qualitative and quantitative analysis, it can be seen that the algorithm proposed in this study has high detection accuracy for small-target water-floating garbage in complex backgrounds.

In addition, in order to verify the effectiveness of the MGA proposed in this study, multi-scale types of attention were added in the same location of YOLOv7 for comparison among PSA attention, MHSA attention (Multi-Head Self Attention) [47], and EMA attention (Efficient Multi-Head Self-Attention) [48], as shown in Table 3.

Table 3 shows that the MGA proposed in this study has certain rationality and superiority, compared with the other three attention mechanisms, in that the mmAP (34.21%) and Recall (83.44%) indexes of MGA are optimal, the mAP ranks third, and the Model Size ranks forth. Because the attention is an extra added network structure, it will cause the Model Size and FPS to decrease. Nevertheless, MGA can still meet the requirements of real-time small-target garbage detection.

### 4.4. Ablation Experiments

To test the effectiveness of APM-YOLOv7 with different improvement strategies, ablation experiments were conducted in the same environment, and the various metrics of the model performance were tested on the test set. The ablation experimental results are shown in Table 4.

As shown in Table 4 and Figure 16, compared to the benchmark YOLOv7, the detection accuracy is further improved after the image is preprocessed with the ACanny river channel outline extraction algorithm, and the FPS and Model Size are the same as those of the benchmark YOLOv7 due to the unaltered network structure. After the PConv-ELAN lightweight network was designed in YOLOv7, all the metrics of the model were compared with those of the benchmark YOLOv7. To be specific, mAP is improved by 2.94%, mmAP by 2.17%, Recall by 7.96%, and FPS by 3.6, and the Model Size is reduced by 9.9 MB, which indicates that this lightweight network structure not only reduces the network’s computational parameters and improves the detection speed, but it also enhances the network’s feature extraction capability for small targets. Adding MGA attention further improves mAP, mmAP, and Recall compared to the benchmark YOLOv7, with mAP being increased by 7.02%, mmAP by 3.91%, Recall by 11.82%, FPS by 2.07, and Model Size by 34.6 MB. Since MGA is an extra added attention mechanism, it will cause FPS and Model Size to be reduced, but it can still meet the requirements of real-time detection in general. Figure 16 shows that after heat map analysis, the addition of the MGA attention mechanism can accurately locate the region of interest in the image and realize highly precise small-target water-floating garbage detection.

## 5. Conclusions

To overcome challenges and difficulties in small-target water-floating garbage detection, an APM-YOLOv7 method is proposed. In order to reduce the interference caused by the complex riverbank background, this study proposes an ACanny river channel outline extraction algorithm, which extracts river features from the complex background by adaptively obtaining the high and low thresholds of the Canny algorithm and connecting the edges of the river using the concave packet algorithm, which makes the network capable of extracting water-floating garbage features more accurately. In order to extract more morphologically variable water-floating garbage features, this study introduces PConv and designs a lightweight multi-scale fusion module of PConv-ELAN, which can improve the detection speed, reduce the computational redundancy, and acquire more morphologically variable water-floating garbage features. Finally, the limitations of the YOLOv7 network in small-target feature extraction are analyzed, and a multi-scale gated attention mechanism, MGA, with adaptive weight allocation is proposed. The MGA uses the idea of “gating” to realize the multi-scale weighted fusion of features extracted from multi-scale convolution by adaptively allocating different weights, highlighting small-target features, and reducing missed detection. The ablation experiments show that compared with the benchmark YOLOv7, the mAP of APM-YOLOv7 is improved by 7.02%, the mmAP by 3.91%, and the Recall by 11.82%, which can satisfy the requirements of high-precision real-time water-floating garbage detection.

However, during the experiment, it was found that the YRDG dataset contains limited scenes, which limited the performance of the APM-YOLOv7 algorithm. In addition, the issue of uneven detection performance for individual categories exists. In the future, we will expand the number and diversity of datasets, including water-floating garbage images in low-light and remote-sensing scenarios. We will also optimize the APM-YOLOv7 algorithm from the aspects of ensemble learning, lightweight model structure design, and multimodal data to enhance its overall performance and real-time detection capability, thereby further expanding its application scenarios and achieving better detection results.

## Figures and Tables

**Figure 1 sensors-24-00050-f001:**
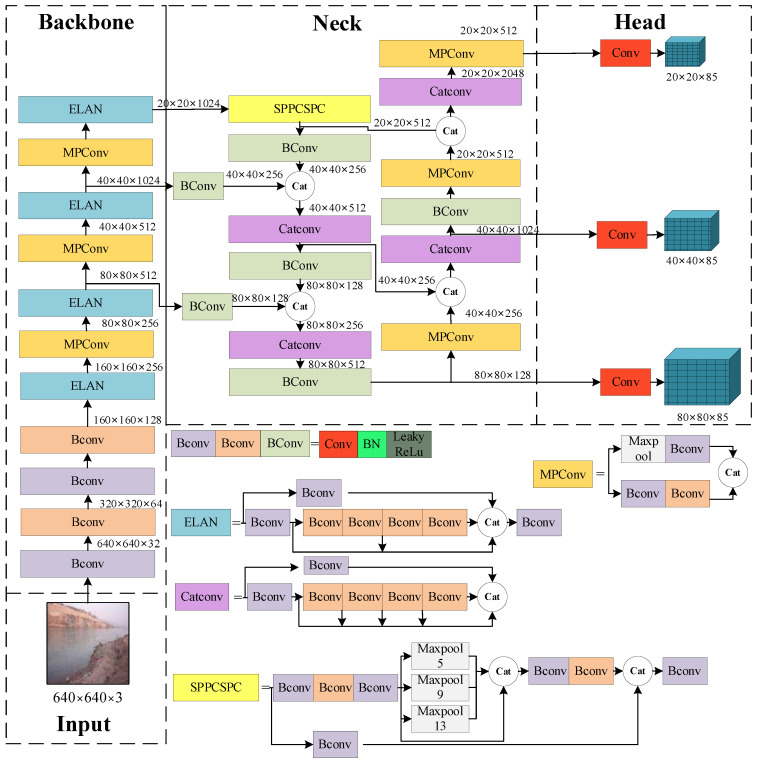
Diagram of YOLOv7 network architecture.

**Figure 2 sensors-24-00050-f002:**
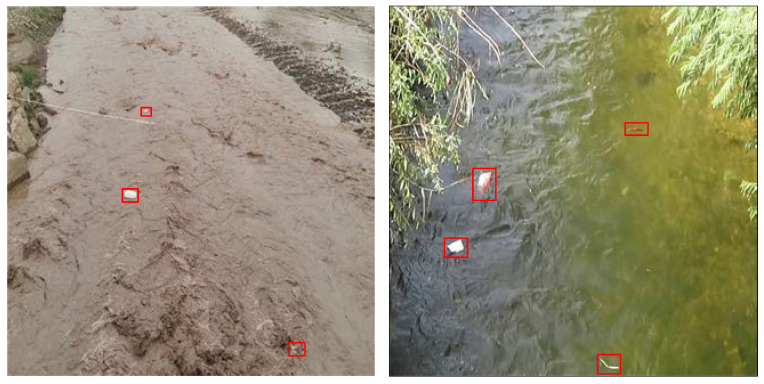
The complex background of small-target water-floating garbage (the target area is framed with a red box).

**Figure 3 sensors-24-00050-f003:**
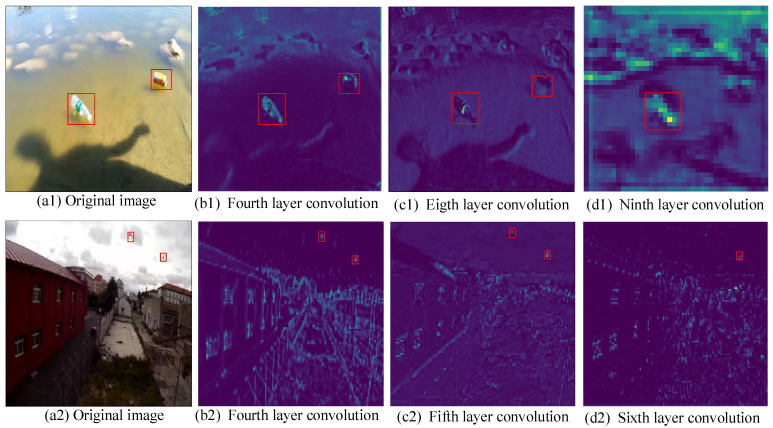
Output feature maps of the different convolutional layers of YOLOv7 (the target area is framed with a red box). (**a1**,**a2**) is the original image, (**b1**,**b2**) and (**c1**,**c2**) are the feature maps extracted by convolution, and (**d1**,**d2**) is the feature map lost by small target features after convolution.

**Figure 4 sensors-24-00050-f004:**
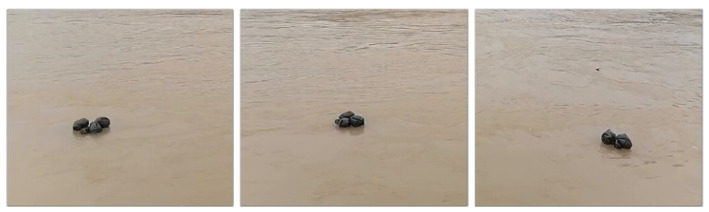
Changes in appearance of the same target.

**Figure 5 sensors-24-00050-f005:**
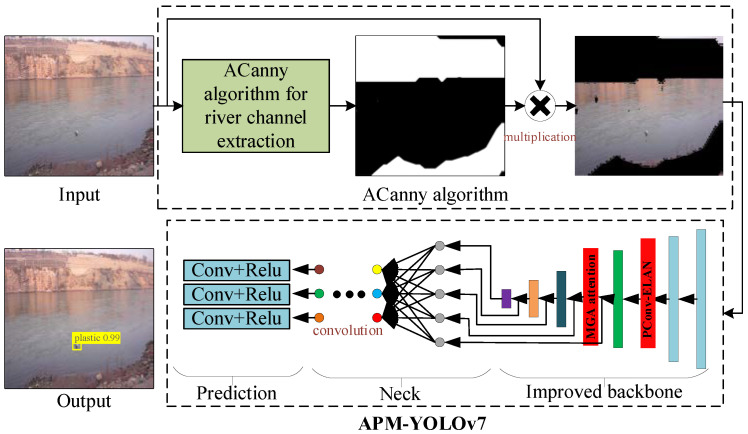
APM-YOLOv7 overall framework.

**Figure 6 sensors-24-00050-f006:**
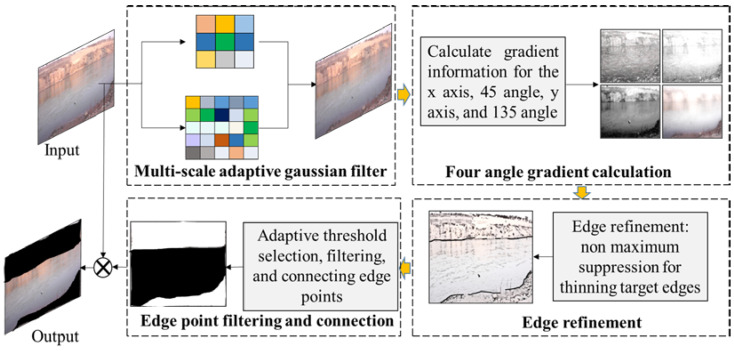
The overall process of the ACanny algorithm.

**Figure 7 sensors-24-00050-f007:**
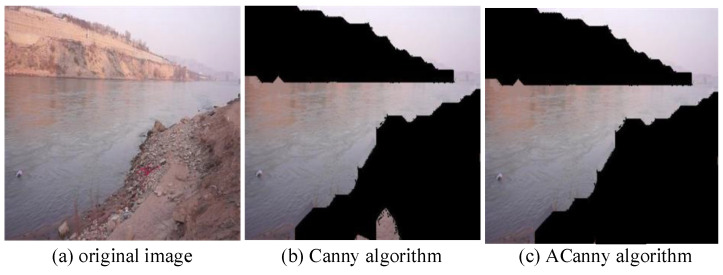
River channel outline extraction results. (**a**) is the original image, (**b**) is the Canny algorithm result graph, and (**c**) is the ACannoy algorithm result graph.

**Figure 8 sensors-24-00050-f008:**
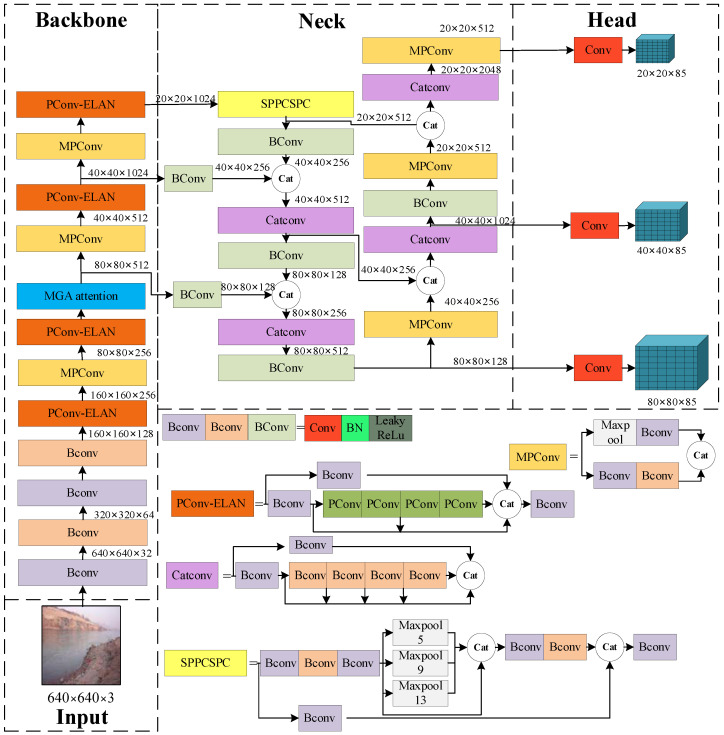
APM-YOLOv7 structure.

**Figure 9 sensors-24-00050-f009:**
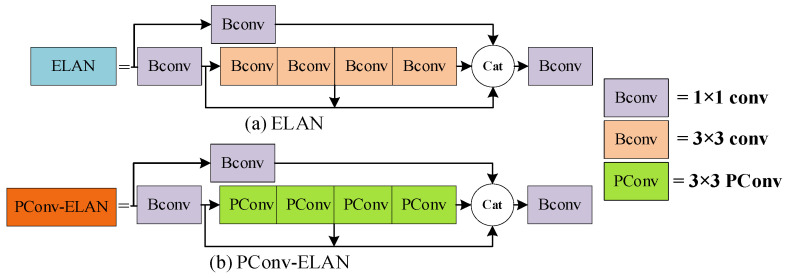
ELAN and PConv-ELAN structures. (**a**) is the ELAN structure, (**b**) is the PConv-ELAN structure.

**Figure 10 sensors-24-00050-f010:**
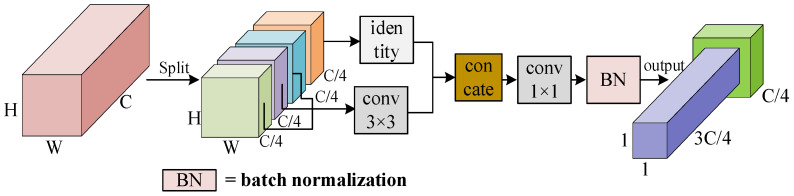
PConv structure.

**Figure 11 sensors-24-00050-f011:**
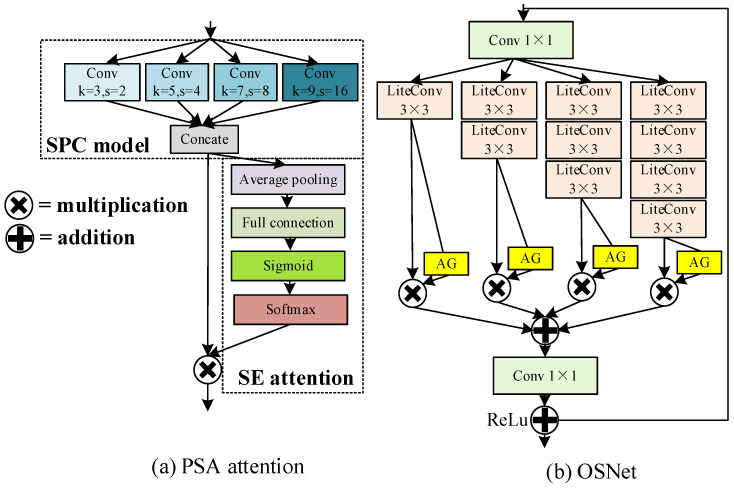
PSA attention and OSNet structure diagrams. (**a**) is PSA attention, (**b**) is OSNet.

**Figure 12 sensors-24-00050-f012:**
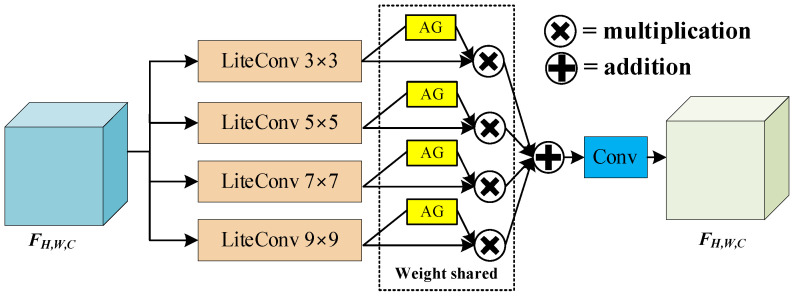
MGA attention structure.

**Figure 13 sensors-24-00050-f013:**
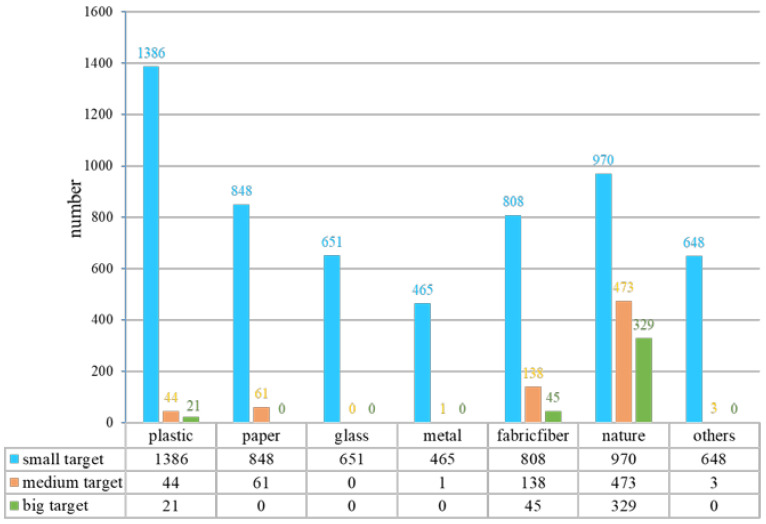
Number of large, medium, and small targets for each category.

**Figure 14 sensors-24-00050-f014:**
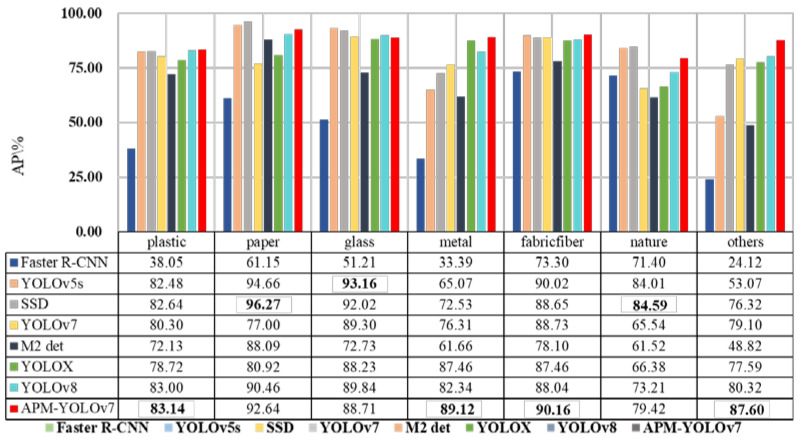
Contrasting algorithms in terms of average AP. (The best indicator for the bold representation algorithm).

**Figure 15 sensors-24-00050-f015:**
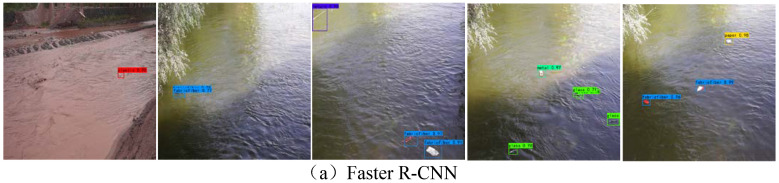
Detection results of different algorithms for small-target water-floating garbage.

**Figure 16 sensors-24-00050-f016:**
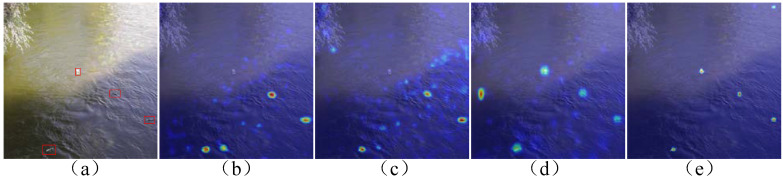
Comparison of heat maps for different improvement strategies (the target area is framed with a red box): (**a**) the original figure, (**b**) the heat map of the YOLOv7 model, (**c**) the heat map of the ACanny + YOLOv7 model, (**d**) the heat map of the ACanny + YOLOv7 + PConv-ELAN model, and (**e**) the heat map of the APM-YOLOv7 model.

**Table 1 sensors-24-00050-t001:** Number of labels and examples in the YRDG dataset.

Serial Number	Categories	Number	Example
1	plastic	1432	Plastic bags and bottles, foam blocks, etc.
2	paper	909	Milk cartons, file folders, etc.
3	glass	651	Glass jars, wine bottles, etc.
4	metal	466	Cans, gasoline drums, etc.
5	fabric/fiber	991	Rope, cloth bags, torn clothes
6	natural	1772	Twigs, wood chips, algae, leaves, etc.
7	others	651	Unrecognizable materials, etc.

**Table 2 sensors-24-00050-t002:** Comparative experiments of different algorithms.

Algorithm	mAP (%)	mmAP (%)	Recall (%)	FPS	Model Size (MB)
Faster R-CNN	50.37	16.82	83.62	16.12	521.0
YOLOv5s	80.35	31.12	82.86	41.36	27.2
SSD	85.15	30.32	40.00	21.80	469.0
YOLOv7	80.30	31.50	72.71	32.57	71.4
M2det	69.01	23.60	29.35	12.00	238.0
YOLOX	79.43	31.87	76.37	**42.84**	34.4
YOLOv8	83.89	32.46	80.59	40.51	75.7
APM-YOLOv7	**87.32**	**35.41**	**84.53**	30.50	106.0

The best indicator for the bold representation algorithm.

**Table 3 sensors-24-00050-t003:** Comparative experiments between different attention models.

Algorithm	Attention Model	mAP (%)	mmAP (%)	Recall (%)	FPS	MS (MB)
YOLOv7	--	80.30	31.50	72.71	32.57	71.4
	PSA	86.81	33.93	78.80	30.40	113.0
	MHSA	80.90	33.32	76.23	28.89	119.5
	EMA	**87.30**	34.20	77.52	30.95	112.1
	MGA	86.61	**34.21**	**83.44**	30.12	115.9

The best indicator for the bold representation algorithm.

**Table 4 sensors-24-00050-t004:** Ablation experimental results with different improvement strategies.

Model	mAP %	mmAP %	Recall %	FPS	Model Size MB
YOLOv7	80.30	31.50	72.71	**32.57**	71.3
ACanny + YOLOv7	82.76	33.36	78.57	**32.57**	71.3
ACanny + YOLOv7 + PConv-ELAN	83.24	33.67	80.67	36.17	**61.4**
ACanny + YOLOv7 + PConv-ELAN + MGA (APM-YOLOv7)	**87.32**	**35.41**	**84.53**	30.50	106.0

The best indicator for the bold representation algorithm.

## Data Availability

The dataset is available at https://github.com/jingcodejing/YRDG-dataset, accessed on 8 October 2023. The code is available at https://github.com/jingcodejing/AAM-attention, accessed on 8 October 2023.

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
