# Peer review of "APM-YOLOv7 for Small-Target Water-Floating Garbage Detection Based on Multi-Scale Feature Adaptive Weighted Fusion"

_sensors, 2023, doi:10.3390/s24010050_

Round 1

Reviewer 1 Report

Comments and Suggestions for Authors

In this paper, a small target detection method based on APM-YOLOv7 is proposed to solve the problem of low detection accuracy due to limited information and complex background of small targets. This work is valuable and interesting. Experiments show that this method is effective and better than some existing methods in terms of mAP, Recall and FPS metrics. Overall, the methodology is impressive. However, there are a couple of concerns I would like to raise:

1) The pictures in the article are too blurred, as shown in Figure 3 and Figure 16, and I can't even see the target and score marks in each picture clearly. It is better to give clear images in the new manuscript.

2) Figures 6 and 10 in the article are too small, so it is suggested that the author enlarge it so as to facilitate observation.

3) The language is colloquial. A review by a fluent English speaker would be welcome. Please carefully check the English description of the whole manuscript.

4) Detection of maritime targets and small targets is a hot topic in the field of remote sensing image processing. It is recommended to cite several valuable articles published recently in the Introduction section, such as:

Infrared Maritime Dim Small Target Detection Based on Spatiotemporal Cues and Directional Morphological Filtering [J], Infrared Physics & Technology, 2021, 115: 103657-103675, 10.1016/j.infrared.2021.103657.

Comments on the Quality of English Language

The language is colloquial. A review by a fluent English speaker would be welcome. Please carefully check the English description of the whole manuscript.

Author Response

Dear Reviewer:

Thank you very much for your valuable feedback on the manuscript we submitted. We attach great importance to your review conclusion and have made revisions based on your suggestions. We have also further polished and revised the language of the manuscript to make it more complete and rigorous. The modifications involve images and references. For ease of reference, we have included the specific modifications in the cover letter.

Reviewer 2 Report

Comments and Suggestions for Authors

1.     In addition to YOLOv7,  there are other mutations or successors, such as YOLO v8, YOLO-nas, YOLOX, and etc. A comparison or some experimental results should be provided to justify the reason that the current version was adopted for the proposed APM-YOLOv7. Especially for YOLO v8 and YOLOX, which was reported to have better performance in different aspect of the literature.

2.     Figure 5 and Figure 6 both have the same caption: “Overall framework” which should be modified accordingly. In addition, the figures should be modified with more information by adding more details about the steps and some more information about the input and output of each step.

3.     According to Figure 15, in addition to the excellent results derived from APM-YOLOv7, YOLOv5s worked better in the category “glass,” and SSD performed better in the categories “paper” and “nature.” it might be a good idea to build an ensemble learning mechanism for the integration of these three algorithms.

4.     Timing results are critical for real-time pattern recognition. The authors have stated the hardware and software environment used for training and testing the small-target water-floating garbage detection; the corresponding timing results should also be reported for comparison.

Author Response

(The authors gave the same response as above.)

Reviewer 3 Report

Comments and Suggestions for Authors

This work introduces a novel small-target detection method, the efficacy of which is validated through experimentation, highlighting its advantages. However, to enhance the overall quality, several key improvements should be addressed:

  1. 1.The subsequent manuscript should include a comprehensive ablation analysis and an exploration of the limitations inherent in this work.

  2. 2.In the revised manuscript, it is imperative to incorporate relevant works for comparison and context. Notable examples include "Learning Semantically Enhanced Features for Fine-grained Image Classification."

  3. 3.The motivation behind this work needs elucidation to facilitate a clearer understanding for readers.

Comments on the Quality of English Language

need to improve

Author Response

(The authors gave the same response as above.)
